# Clinical severity of SARS-CoV-2 Omicron BA.4 and BA.5 lineages compared to BA.1 and Delta in South Africa

Nicole Wolter [1,2] ✉, Waasila Jassat [3,4], Sibongile Walaza[1,5], Richard Welch [3,4], Harry Moultrie [2,6], Michelle J. Groome [2,3], Daniel Gyamfi Amoako [1,7], Josie Everatt [1], Jinal N. Bhiman [8,9], Cathrine Scheepers [8,9], Naume Tebeila[1], Nicola Chiwandire[1], Mignon du Plessis [1,2], Nevashan Govender[3], Arshad Ismail [10,11], Allison Glass[2,12], Koleka Mlisana[2,13,14], Wendy Stevens[2,13], Florette K. Treurnicht[2,13], Kathleen Subramoney[2,13], Zinhle Makatini[2,13], Nei-yuan Hsiao [13,15], Raveen Parboosing[2,13,16], Jeannette Wadula[2,13,17], Hannah Hussey[18], Mary-Ann Davies[18], Andrew Boulle[18], Anne von Gottberg [1,2,19] & Cheryl Cohen [1,5,19]

Omicron lineages BA.4 and BA.5 drove a fifth wave of COVID-19 cases in South Africa. Here, we use the presence/absence of the S-gene target as a proxy for SARS-CoV-2 variant/lineage for infections diagnosed using the TaqPath PCR assay between 1 October 2021 and 26 April 2022. We link national COVID-19 individual-level data including case, laboratory test and hospitalisation data. We assess severity using multivariable logistic regression comparing the risk of hospitalisation and risk of severe disease, once hospitalised, for Delta, BA.1, BA.2 and BA.4/BA.5 infections. After controlling for factors associated with hospitalisation and severe outcome respectively, BA.4/BA.5-infected individuals had a similar odds of hospitalisation (aOR 1.24, 95% CI 0.98–1.55) and severe outcome (aOR 0.72, 95% CI 0.41–1.26) compared to BA.1-infected individuals. Newly emerged Omicron lineages BA.4/BA.5 showed similar severity to the BA.1 lineage and continued to show reduced clinical severity compared to the Delta variant.

The Omicron SARS-CoV-2 variant of concern (VOC) was first detected in South Africa in mid-November 2021, with the BA.1 lineage driving a fourth wave of infections in the country[1]. The BA.2 lineage became dominant in the period when SARS-CoV-2 case numbers were declining from the fourth wave peak, resulting in a slower decline and higher baseline than observed in previous inter-wave periods. In week 1 of 2022 BA.1 comprised most sequences (72%), however was replaced by BA.2 during the first few weeks of 2022 which constituted 66% of sequences in week 4. New lineages of Omicron (BA.4 and BA.5) were detected by the Network for Genomic Surveillance in South Africa

(NGS-SA) in January and February 2022, respectively, and by April were the dominant lineages in the country[2]. BA.4 and BA.5 lineages were responsible for a fifth wave of infections, and varied geographically in dominance between the two lineages. As of end April 2022, 49.6% of individuals aged ≥18 years in South Africa had received at least one dose of COVID-19 vaccine (BNT162b or Ad26.COV2.S): 36.2%, 53.4%, 65.7% and 70.7% in the 18–34 years, 35–49 years, 50–59 years and ≥60 years age groups respectively[3].

At the time of this study, BA.4 and BA.5 have identical spike proteins, with additional mutations in the NTD (69–70 deletion) and RBD

---

**Table 1 | Characteristics of individuals infected with SARS-CoV-2 by variant/lineage type, 1 October – 26 April 2021[a] (N = 98,710)**

| | Delta[b] n (%) | BA.1[b] n (%) | BA.2[b] n (%) | BA.4/BA.5[b] n (%) |
|---|---|---|---|---|
| **Age group (years)** | | | | |
| | N = 1273 | N = 75,563 | N = 20,068 | N = 1806 |
| <5 | 19 (1) | 1267 (2) | 517 (3) | 30 (2) |
| 5–12 | 57 (4) | 3071 (4) | 2369 (12) | 134 (7) |
| 13–18 | 128 (10) | 3809 (5) | 2590 (13) | 105 (6) |
| 19–24 | 99 (8) | 5782 (8) | 1255 (6) | 116 (6) |
| 25–39 | 394 (31) | 28,371 (38) | 5264 (26) | 581 (32) |
| 40–59 | 372 (29) | 24,624 (33) | 5889 (29) | 591 (33) |
| ≥60 | 204 (16) | 8639 (11) | 2184 (11) | 249 (14) |
| **Sex** | | | | |
| | N = 1255 | N = 74,692 | N = 19,984 | N = 1795 |
| Male | 577 (46) | 32,780 (44) | 9086 (45) | 833 (46) |
| Female | 678 (54) | 41912 (56) | 10,898 (55) | 962 (54) |
| **Province** | | | | |
| | N = 1243 | N = 73,921 | N = 19,948 | N = 1790 |
| Eastern Cape | 0 (0) | 86 (0) | 9 (0) | 1 (0) |
| Free State | 67 (5) | 2132 (3) | 118 (1) | 4 (0) |
| Gauteng | 44 (37) | 38,945 (53) | 11,935 (60) | 1141 (64) |
| KwaZulu-Natal | 398 (32) | 16,680 (23) | 3701 (19) | 563 (31) |
| Limpopo | 26 (2) | 2758 (4) | 504 (3) | 6 (0) |
| Mpumalanga | 44 (4) | 3724 (5) | 1862 (9) | 28 (2) |
| North West | 35 (3) | 3350 (5) | 691 (3) | 15 (1) |
| Northern Cape | 67 (5) | 1192 (2) | 41 (0) | 1 (0) |
| Western Cape | 152 (12) | 5054 (7) | 1087 (5) | 31 (2) |
| **Hospital admission[c]** | | | | |
| | N = 1273 | N = 75,563 | N = 20,068 | N = 1806 |
| No | 1101 (86) | 72,553 (96) | 19,405 (97) | 1719 (95) |
| Yes | 172 (14) | 3010 (4) | 663 (3) | 87 (5) |
| **Healthcare sector** | | | | |
| | N = 1273 | N = 75,563 | N = 20,068 | N = 1806 |
| Public | 635 (50) | 24,760 (33) | 2025 (10) | 180 (10) |
| Private | 638 (50) | 50,803 (67) | 18,043 (90) | 1626 (90) |
| **Re-infection[d]** | | | | |
| | N = 1273 | N = 75,563 | N = 20,068 | N = 1806 |
| No | 12,336 (97) | 68,227 (90) | 18,202 (91) | 1594 (88) |
| Yes | 37 (3) | 7336 (10) | 1866 (9) | 212 (12) |
| **Co-morbidity[e,f]** | | | | |
| | N = 168 | N = 2940 | N = 637 | N = 80 |
| No | 100 (60) | 2135 (73) | 454 (71) | 46 (58) |
| Yes | 68 (40) | 805 (27) | 183 (29) | 34 (43) |
| **COVID-19 Vaccination[f,g]** | | | | |
| | N = 168 | N = 2940 | N = 637 | N = 80 |
| No | 48 (29) | 932 (32) | 203 (32) | 12 (15) |
| Yes | 2 (1) | 143 (5) | 37 (6) | 3 (4) |
| Unknown | 118 (70) | 1865 (63) | 397 (62) | 65 (81) |

[a]Cases only include individuals whose infection was diagnosed using the TaqPath PCR assay. Individuals were followed-up for outcome until 11 May 2022.

[b]SGTP infections diagnosed in October and November 2021 were classified as Delta, SGTF infections diagnosed between November 2021 through January 2022 were classified as BA.1, SGTP infections diagnosed from February through April 2022 were classified as BA.2 and SGTF infections diagnosed in April 2022 were classified as BA.4/BA.5

[c]Admission to hospital between 7 days prior to 21 days after diagnosis (specimen collection date).

[d]Re-infection was defined as an individual with at least one positive SARS-CoV-2 test >90 days prior to the current episode.

[e]Co-morbidity defined as ≥1 of the following conditions: hypertension, diabetes, chronic cardiac disease, chronic kidney disease, asthma, chronic obstructive pulmonary disease, malignancy, HIV, and active or past tuberculosis.

[f]Only available for hospitalized patients.

[g]Vaccination defined as at least one dose of Ad26.COV2.S or two doses of BNT162b

(L452R, F486V and wild-type Q493) regions compared to BA.2[2]. These new lineages were estimated to have a growth advantage over BA.2, and showed reduced neutralization by serum obtained from individuals that had received three doses of COVID-19 vaccine (BNT162b or ChAdOx1-S) compared to BA.1 and BA.2[2,4]. In addition, BA.4 and BA.5 have shown escape from BA.1 elicited immunity[5]. An increasing number of countries have reported detection and increasing prevalence of BA.4 and/or BA.5, despite prior waves of Omicron BA.1 and BA.2.

We previously reported that Omicron BA.1 was associated with a lower risk of hospitalisation and lower risk of severe illness, compared to Delta variant infection[6]. A similar proportion of individuals were hospitalised and developed severe illness when infected with BA.2 compared to BA.1[7]. Reduced severity of BA.1 and BA.2 infections was also observed in other parts of the world[8,9]. The BA.1, BA.4 and BA.5 lineages are associated with S-gene target failure (SGTF) when tested using the TaqPath™ COVID-19 PCR test (Thermo Fisher Scientific, Waltham, MA, USA) due to the 69–70 position amino acid deletion in the spike protein. Omicron BA.2 lacks this deletion, and therefore BA.2 infections are S-gene positive on this assay. It is important to understand clinical severity associated with infections due to BA.4 and BA.5, as the prevalence of these new lineages increases in other parts of the world. Leveraging on the presence or absence of the S-gene target, we aimed to assess the severity of BA.4/BA.5 lineages in the South African population.

## Results

In the period 1 October 2021 through 26 April 2022, 884,379 COVID-19 cases were diagnosed. Of these, 144,086 (16.3%) were known to have been diagnosed using the TaqPath™ COVID-19 PCR test (Supplementary figure 1), ranging by province from 3.7% to 23.6% of cases (Supplementary table 2). In the public sector 11.1% (43,752/395,924) of cases were diagnosed with the TaqPath test, and 20.5% (100,334/488,455) in the private sector (P < 0.001). The proportion of SGTP and SGTF infections varied throughout the period, with alternating dominance. Using the time period together with the presence/absence of the S-gene target, 98,710 infections could be classified as the likely SARS-CoV-2 variant/lineage: 1273 (1.3%) Delta; 75,563 (76.6%) BA.1; 20,086 (20.3%) BA.2; 1,806 (1.8%) BA.4/BA.5 infections. The median age and interquartile range of cases with known variant/lineage was 37 (26–49) years. Females accounted for 55.7% (54,450/97,726) of cases.

Characteristics of individuals by variant/lineage are shown in Table 1. The median age and interquartile range of cases for each variant was 37 (25–52) years, 37 (28–49) years, 35 (17–48) years and 38 (27–51) years for Delta, BA.1, BA.2 and BA.4/BA.5 respectively. A larger proportion of cases with BA.2 infections were children aged 5 to 18 years (24.7% BA.2 compared to 14.5% Delta, 9.1% BA.1 and 13.2% BA.4/BA.5). Among individuals with Delta infections, 13.5% were admitted to hospital, compared to 4.0% BA.1, 3.3% BA.2 and 4.8% BA.4/BA.5 (Table 1). The proportion of infections identified as re-infections was higher for all Omicron-infected individuals (9.7% BA.1, 9.3% BA.2 and 11.7% BA.4/BA.5) compared to Delta-infected individuals (2.9%). Self-reported vaccination status was only available for 37.5% (22,734/60,662) of hospitalised individuals, of which 13.9% (3157/22,734) reported being fully vaccinated (at least one dose of Ad26.COV2.S or two doses of BNT162b); 4.0%, 13.3%, 15.4% and 20.0% with Delta, BA.1, BA.2 and BA.4/BA.5 infections, respectively.

On multivariable analysis, after controlling for factors associated with hospitalisation and compared to BA.1 infection, the odds of being admitted to hospital was higher for Delta-infected individuals (adjusted odds ratio (aOR) 3.41, 95% confidence interval (CI) 2.86–4.07), lower for BA.2-infected individuals (aOR 0.90, 95% CI 0.82–0.98), and was not significantly different for BA.4/BA.5-infected individuals (aOR 1.24, 95% CI 0.98–1.55) (Table 2). In addition to geographic factors, hospital admission was associated with young age (<5 years, aOR 7.09, 95% CI 5.81–8.67) and older age (40–59 years, aOR 1.36, 95%CI 1.15–1.60 and ≥60 years, aOR 4.77, 95% CI 4.06–5.60) compared to individuals aged 19–24 years, and female sex (aOR 1.09, 95%CI 1.02–1.17). Individuals were less likely to be admitted to hospital in the private sector (aOR 0.58, 95% 0.54–0.63) compared to the public sector. Using this same model, with Delta variant infections as the

**Table 2 | Multivariable logistic regression analysis evaluating the association between SARS-CoV-2 variant/lineage and hospitalisation, South Africa, 1 October 2021 – 26 April 2022[a] (N = 95,940)**

| | | Hospital admission[b] n/N (%) | Odds ratio (95% CI) | Adjusted odds ratio (95% CI) | P value |
|---|---|---|---|---|---|
| SARS-CoV-2 variant/lineage[c] | | N = 98,710 | | | |
| | Delta | 172/1273 (14) | 3.77 (3.19–4.44) | 3.41 (2.86–4.07) | <0.001 |
| | BA.1 | 3010/75,763 (4) | Ref | Ref | – |
| | BA.2 | 663/20,068 (3) | 0.82 (0.76–0.90) | 0.90 (0.82–0.98) | 0.021 |
| | BA.4/BA.5 | 87/1806 (5) | 1.22 (0.98–1.52) | 1.24 (0.98–1.55) | 0.070 |
| Age group (years) | | N = 98,710 | | | |
| | <5 | 271/1833 (15) | 6.66 (5.48–8.10) | 7.09 (5.81–8.66) | <0.001 |
| | 5–12 | 139/5631 (2) | 0.97 (0.78–1.22) | 1.13 (0.90–1.42) | 0.280 |
| | 13–18 | 139/6632 (2) | 0.82 (0.66–1.03) | 0.91 (0.72–1.14) | 0.402 |
| | 19–24 | 184/7252 (3) | Ref | Ref | – |
| | 25–39 | 995/34,610 (3) | 1.14 (0.97–1.33) | 1.16 (0.98–1.36) | 0.077 |
| | 40–59 | 1008/31,476 (3) | 1.27 (1.08–1.49) | 1.36 (1.15–1.60) | <0.001 |
| | ≥60 | 1196/11,276 (11) | 4.56 (3.89–5.34) | 4.77 (4.06–5.60) | <0.001 |
| Sex | | N = 97,726 | | | |
| | Male | 1663/43,276 (4) | Ref | Ref | – |
| | Female | 2249/54,450 (4) | 1.08 (1.01–1.15) | 1.09 (1.02–1.17) | 0.009 |
| Province | | N = 96,902 | | | |
| | Eastern Cape | 4/96 (4) | 1.75 (0.64–4.83) | 1.87 (0.67–5.21) | 0.233 |
| | Free State | 112/2321 (5) | 2.04 (1.59–2.62) | 1.55 (1.20–2.00) | 0.001 |
| | Gauteng | 1847/52,475 (4) | 1.47 (1.25–1.74) | 1.54 (1.30–1.82) | <0.001 |
| | KwaZulu-Natal | 1146/21,342 (5) | 2.29 (1.93–2.72) | 2.16 (1.81–2.57) | <0.001 |
| | Limpopo | 99/3294 (3) | 1.25 (0.97–1.62) | 1.64 (1.26–2.12) | <0.001 |
| | Mpumalanga | 237/5658 (4) | 1.76 (1.43–2.17) | 2.26 (1.83–2.79) | <0.001 |
| | North West | 190/4091 (5) | 1.96 (1.58–2.44) | 2.39 (1.92–2.98) | <0.001 |
| | Northern Cape | 41/1301 (3) | 1.31 (0.92–1.86) | 0.92 (0.64–1.32) | 0.652 |
| | Western Cape | 153/6324 (2) | Ref | Ref | - |
| Healthcare sector | | N = 98,710 | | | |
| | Public | 1499/27,600 (5) | Ref | Ref | - |
| | Private | 2433/71,110 (3) | 0.62 (0.58–0.66) | 0.58 (0.54–0.63) | <0.001 |

[a]Individuals followed-up for hospital admission until 11 May 2022.
[b]Admission to hospital between 7 days prior to 21 days after diagnosis (specimen collection date).
[c]SGTP infections diagnosed in October and November 2021 were classified as Delta, SGTF infections diagnosed between November 2021 through January 2022 were classified as BA.1, SGTP infections diagnosed from February through April 2022 were classified as BA.2 and SGTF infections diagnosed in April 2022 were classified as BA.4/BA.5.

reference group, all Omicron lineages showed a reduced odds of hospitalisation (BA.1, aOR 0.29, 95% CI 0.25–0.35; BA.2, aOR 0.26, 95% CI 0.22–0.32; BA.4/BA.5, aOR 0.36, 95% CI 0.27–0.48). In the sensitivity analysis for risk of hospitalisation, individuals infected with BA.4/BA.5 had similar odds of being admitted compared to individuals infected with BA.2 (aOR 1.00, 95% CI 0.65–1.54).

Among the 98,710 SARS-CoV-2 infected individuals with variant/lineage assigned and known outcome, 3825 (3.9%) were admitted to hospital. Among admitted individuals with known outcome, 1276 (33.4%) developed severe disease: 57.7% (97/168) with Delta, 33.7% (990/2940) with BA.1, 26.2% (167/637) with BA.2 and 27.5% (22/80) with BA.4/BA.5 infections. On multivariable analysis compared to BA.1 infection, the odds of severe disease was only higher for Delta infection (aOR 2.47, 95% CI 1.73–3.52), was lower for BA.2 infection (aOR 0.78, 95% CI 0.63–0.97) and did not differ for BA.4/BA.5 infection (aOR 0.72, 95% CI 0.41–1.26) (Table 3). The odds of severe disease was higher among individuals aged 40–59 years (aOR 2.57, 95% CI 1.65–4.00) and ≥60 years (aOR 5.22, 95% CI 3.37–8.08) compared to individuals aged 19–24 years, as well as among individuals with underlying illness (aOR1.57, 95% CI 1.32–1.87). The odds of severe disease was lower for females (aOR 0.81, 95% CI 0.69–0.94) and individuals hospitalised in the private sector (aOR 0.72, 95% CI 0.60–0.86). The odds of severe disease did not differ for individuals that had been fully vaccinated (aOR 0.80, 95% CI 0.54–1.17), likely due to low numbers. Using this

same model, with Delta variant infections as the reference group, all Omicron lineages showed a reduced odds of severe disease (BA.1, aOR 0.41, 95% CI 0.28–0.58; BA.2, aOR 0.32, 95% CI 0.21–0.47; BA.4/BA.5, aOR 0.29, 95% CI 0.15–0.56). In the sensitivity analysis for risk of severe disease, individuals infected with BA.4/BA.5 had similar odds of severe disease compared to individuals infected with BA.2 (aOR 0.75, 95% CI 0.36–1.58).

## Discussion

Omicron lineages have continued to emerge, most recently with the detection of BA.4 and BA.5 lineages[2]. Rapidly characterising the new variants and lineages, and the effect on case numbers and thereby the impact on the healthcare system, is important. In this study, we aimed to determine the severity of Omicron BA.4 and BA.5 infections in South Africa through analysis of the risk of hospitalisation, and severe outcome once hospitalised. We found that BA.4/BA.5 infected individuals were not more likely to be hospitalised or develop severe disease compared to BA.1. Similarly to what was previously described[6], individuals infected with Delta variant showed increased severity (risk of hospitalisation and severe disease) compared to BA.1.

During the fourth COVID-19 wave in South Africa, driven by Omicron BA.1, the country observed a de-coupling of the incidence of COVID-19 hospitalisations and deaths from incidence of infections[10].

**Table 3 | Multivariable logistic regression analysis evaluating the association between SARS-CoV-2 variant/lineage and severe disease among hospitalised individuals, South Africa, 1 October 2021 – 26 April 2022[a] (N = 3574)**

| | | Severe disease[b] n/N (%) | Odds ratio (95% CI) | Adjusted odds ratio (95% CI) | P value |
|---|---|---|---|---|---|
| SARS-CoV-2 variant/lineage[c] | | N = 3825 | | | |
| | Delta | 97/168 (58) | 2.69 (1.96–3.69) | 2.47 (1.73–3.52) | <0.001 |
| | BA.1 | 990/2940 (34) | Ref | Ref | - |
| | BA.2 | 167/637 (26) | 0.70 (0.58–0.85) | 0.78 (0.63–0.97) | 0.029 |
| | BA.4/BA.5 | 22/80 (28) | 0.75 (0.45–1.23) | 0.72 (0.41–1.26) | 0.252 |
| Age group (years) | | N = 3825 | | | |
| | <5 | 45/267 (17) | 0.94 (0.57–1.54) | 1.03 (0.60–1.76) | 0.922 |
| | 5–12 | 11/137 (8) | 0.40 (0.20–0.83) | 0.48 (0.23–1.02) | 0.058 |
| | 13–18 | 18/134 (13) | 0.72 (0.38–1.34) | 0.81 (0.42–1.58) | 0.542 |
| | 19–24 | 32/180 (18) | Ref | Ref | – |
| | 25–39 | 188/976 (19) | 1.10 (0.73–1.67) | 1.06 (0.68–1.66) | 0.795 |
| | 40–59 | 351/968 (36) | 2.63 (1.76–3.94) | 2.57 (1.65–4.00) | <0.001 |
| | ≥60 | 631/1163 (54) | 5.49 (3.68–8.18) | 5.22 (3.37–8.08) | <0.001 |
| Sex | | N = 3808 | | | |
| | Male | 600/1616 (37) | Ref | Ref | – |
| | Female | 669/2192 (31) | 0.74 (0.65–0.85) | 0.81 (0.69–0.94) | 0.007 |
| Province | | N = 3727 | | | |
| | Eastern Cape | 1/4 (25) | 1.06 (0.11–10.56) | 2.48 (0.20–30.12) | 0.475 |
| | Free State | 43/99 (43) | 2.45 (1.42–4.23) | 3.80 (2.01–7.16) | <0.001 |
| | Gauteng | 650/1800 (36) | 1.81 (1.23–2.66) | 3.31 (2.10–5.21) | <0.001 |
| | KwaZulu-Natal | 322/1110 (29) | 1.31 (0.88–1.94) | 2.15 (1.34–3.43) | 0.001 |
| | Limpopo | 19/98 (19) | 0.77 (0.41–1.44) | 1.62 (0.79–3.30) | 0.188 |
| | Mpumalanga | 76/237 (32) | 1.51 (0.95–2.40) | 2.67 (1.53–4.64) | 0.001 |
| | North West | 51/189 (27) | 1.18 (0.72–1.93) | 2.46 (1.39–4.37) | 0.002 |
| | Northern Cape | 31/39 (79) | 12.38 (5.22–29.33) | 11.66 (4.59–29.61) | <0.001 |
| | Western Cape | 36/151 (24) | Ref | Ref | - |
| Co-morbidity[d] | | N = 3825 | | | |
| | Absent | 772/2735 (28) | Ref | Ref | - |
| | Present | 504/1090 (46) | 2.19 (1.89–2.53) | 1.57 (1.32–1.87) | <0.001 |
| Healthcare sector | | N = 3825 | | | |
| | Public | 559/1436 (39) | Ref | Ref | – |
| | Private | 717/2389 (30) | 0.67 (0.59–0.77) | 0.72 (0.60–0.86) | <0.001 |
| Days between diagnosis and admission | | N = 3673 | | | |
| | 1–7 days before diagnosis | 129/340 (38) | Ref | Ref | - |
| | 0–6 days after diagnosis | 969/3023 (32) | 0.77 (0.61–0.97) | 0.84 (0.65–1.10) | 0.206 |
| | 7–21 days after diagnosis | 112/310 (36) | 0.93 (0.67–1.27) | 0.96 (0.67–1.38) | 0.826 |
| SARS-CoV-2 vaccination[e] | | N = 3825 | | | |
| | No | 403/1195 (34) | Ref | Ref | - |
| | Yes | 54/185 (29) | 0.81 (0.58–1.14) | 0.80 (0.54–1.17) | 0.250 |
| | Unknown | 819/2445 (34) | 0.99 (0.86–1.15) | 0.92 (0.77–1.09) | 0.339 |

[a]Individuals followed-up for in-hospital outcome until 11 May 2022.
[b]Severe disease defined as a hospitalised patient meeting at least one of the following criteria: admitted to ICU, received oxygen treatment, ventilated, received extracorporeal membrane oxygenation (ECMO), experienced acute respiratory distress syndrome (ARDS) and/or died.
[c]SGTP infections diagnosed in October and November 2021 were classified as Delta, SGTF infections diagnosed between November 2021 through January 2022 were classified as BA.1, SGTP infections diagnosed from February through April 2022 were classified as BA.2 and SGTF infections diagnosed in April 2022 were classified as BA.4/BA.5.
[d]Co-morbidity defined as ≥1 of the following conditions: hypertension, diabetes, chronic cardiac disease, chronic kidney disease, asthma, chronic obstructive pulmonary disease (COPD), malignancy, HIV, and active or past tuberculosis.
[e]Vaccination defined as at least one dose of Ad26.COV2.S or two doses of BNT162b.

Much of this reduced severity is thought to be due to the high population immunity due to previous infection and/or vaccination in the South African population, which prior to the Omicron BA.1 wave was 73%[10]. Although the large number of mutations in the spike protein of Omicron facilitates increased immune escape, the cellular immune response elicited by infection and/or vaccination recognises the Omicron variant[11] and likely protects individuals with BA.4/BA.5 infection against severe disease.

In our study, among the group of individuals infected with BA.4/BA.5 lineages we did not observe increased clinical severity compared to the group of individuals infected with BA.1. Recent data from a study conducted in hamsters showed that BA.4/BA.5 was more pathogenic than BA.2[12]. However, in our study, this increased pathogenicity did not translate into clinical severity. Recent national seroprevalence data among healthy blood donors post the BA.1 fourth wave in South Africa showed that 97% of individuals had SARS-CoV-2 antibodies, with 87%

due to prior infection and an additional 10% due to vaccination alone[13]. This extremely high population immunity, and specifically the elicited T-cell responses, could explain the continuing low severity observed despite the emergence of additional Omicron lineages with increased transmissibility and immune escape. Our findings are supported by the low number of COVID-19 hospitalisations and in-hospital deaths reported during the fifth wave that was driven by the BA.4/BA.5 lineages[14]. However, this may not translate into reduced severity in other settings that do not have high population immunity as the increased transmissibility and immune escape of BA.4/BA.5 may lead to an increase in infections and thereby an increase in hospitalisations, with implications for healthcare systems.

Among the group of individuals with BA.2 infection, we observed a slightly reduced risk of hospitalisation and severe disease compared to those infected with BA.1, although bordering on statistical significance. We previously reported that a similar proportion of individuals were hospitalised and developed severe illness for individuals infected with BA.1 compared to BA.2, although at an earlier point towards the end of the fourth wave in January 2022[7]. This may be because the BA.2 lineage became dominant in the period when SARS-CoV-2 case numbers were declining from the fourth wave (BA.1) peak and population immunity was high with high numbers of infections and re-infections during this wave[15].

During the study period ranging from the third (Delta) wave through the fifth (BA.4/BA.5) wave, there was a shift in cases over time to predominate in the private sector. While the reason for this could be influenced by a number of differences between the public and private sectors, it may reflect reduced COVID-19 testing among individuals and clinicians in the public sector as testing protocols shifted to hospitalised individuals, whereas individuals in the private sector would have had consistent access to testing even when the need for COVID-19 diagnosis for public health interventions and containment became less essential.

Our study has several limitations. The analysis is restricted to infections that were diagnosed using the TaqPath™ COVID-19 PCR assay, which was not used to the same extent throughout the country and therefore may bias the data geographically. Additionally, it is possible that individuals with more severe disease were more likely to have been diagnosed by PCR than antigen tests, resulting in potential bias of our study population towards more severe disease. We used the combination of time period and presence/absence of the S-gene target as a proxy for variant/lineage. There may therefore have been some misclassification of the variants. To limit misclassification, we selected time periods based on sequencing data generated by NGS-SA[1]. Vaccination information was restricted to hospitalised cases and was based on self-report, and as a result the analysis of severe disease among hospitalised individuals was likely more robust than the hospitalisation analysis. We were not able to control for prior SARS-CoV-2 infection as only less than 10% of SARS-CoV-2 infections are diagnosed[16]. This resulted in not adjusting for the effects of previous infection in the multivariable models and therefore some of the observed effect of lower severity in the BA.4/BA.5 wave compared to pre-Omicron waves may be as a result of immunity from prior infection rather than reduced intrinsic virulence. Similarly, if a substantially higher proportion of individuals infected with BA.4/BA.5 had undiagnosed infection compared to BA.1 this may have led to a false impression of equal severity when in fact the intrinsic severity of BA.4/BA.5 could be somewhat higher than that of BA.1. Analysis of the proportion of individuals hospitalised could be affected by changes in testing practices. During the BA.4/BA.5 wave in some provinces there was a shift to preferential testing of hospitalised individuals[17], which would have biased our study population of individuals tested for SARS-CoV-2 to more severe disease and would therefore make our estimate for risk of hospitalisation of BA.4/BA.5 infected individuals a minimum estimate. In addition, we compared infections throughout the full Omicron BA.1

wave to infections in the ascending phase of the Omicron BA.4/BA.5 wave, this could bias comparisons if case characteristics differ in the ascending and descending wave phases or if threshold for hospitalisation changed in these time periods. While the earlier phase of a wave may affect the younger and healthy population groups first before reaching the more vulnerable populations, previous data from DAT-COV has indicated the proportion of severe cases does not vary substantially through the different wave periods[18]. We did not have data on body mass index, a known risk factor for severe COVID-19[19], and could therefore not include this variable in our analysis. Lastly, DATCOV surveillance includes individuals with a SARS-CoV-2 positive test that may have been hospitalised with COVID-19 symptoms or for other non-COVID-19 related conditions which may have overestimated the number of hospitalisations. While DATCOV surveillance contains a field in the web-based platform to indicate if a person was admitted for COVID-19 symptoms or for another reason, the submitted data for this field was often incomplete. As a result, data on reason for admission is missing for approximately 60% of patients. Among those for whom data was available, the proportion of patients admitted for COVID-19 symptoms was 75% (first wave), 78% (second wave), 76% (third wave), 70% (fourth wave) and 74% (fifth wave). In addition, the DATCOV case definition was consistent throughout the study period, and would have affected each time period in the analysis consistently.

We found that in South Africa, where almost the entire population has SARS-CoV-2 antibodies, individuals infected with BA.4/BA.5 had a similar risk of hospitalisation and developing severe disease to individuals with BA.1 infection. Despite the emergence of BA.4/BA.5 leading to a fifth resurgence of cases in the country, data from early in the wave indicate that this may not translate into severity levels observed in waves prior to Omicron when population immunity was lower. In addition, all Omicron lineages analysed (BA.1, BA.2, BA.4/BA.5) showed reduced severity compared to the Delta variant. As the prevalence of cases due to BA.4/BA.5 lineages increase in other countries, this data may be useful for healthcare resource planning, however our findings may not be fully extrapolated to other settings with different immune landscapes such as those with a higher proportion of immunity due to vaccination and not previous infection.

## Methods

We performed a data linkage study, using methods that have been previously described in detail[6]. Briefly, we linked national individual-level data from three sources: (i) national COVID-19 case data, (ii) SARS-CoV-2 laboratory test data for public sector laboratories and one large private sector laboratory, and (iii) DATCOV, which is an active surveillance system for COVID-19 hospital admissions in South Africa. The national COVID-19 case database is a laboratory-based surveillance programme which receives real-time electronic data on all laboratory-confirmed SARS-CoV-2 cases in South Africa. DATCOV contains data for all individuals with a positive PCR or antigen test, with a confirmed duration of stay in hospital of ≥1 day, regardless of age or reason for admission. This included patients with COVID-19 symptoms, acquired nosocomial COVID-19 infection, or tested positive incidentally when admitted for other reasons. DATCOV data were submitted through an electronic (web-based) platform and stored in a Microsoft Azure SQL database. Case and test data were obtained on 26 April 2022, and DATCOV data on 11 May 2022. The dataset was restricted to tests performed on the TaqPath™ COVID-19 assay (Thermo Fisher Scientific, Waltham, MA, USA). We used a combination of the presence (S-gene target positive, SGTP) or absence (S-gene target failure, SGTF) of the S-gene together with time period of circulating variants/lineages from genomic surveillance in South Africa (Supplementary table 1), as a proxy for the variant and lineage[1]. We restricted to tests with a Ct value ≤30 for either the ORF1ab or nucleocapsid (N) gene targets to avoid incorrectly classifying infections as SGTF for which S gene was not detected because of low viral load. SGTP infections diagnosed in

October and November 2021 were classified as Delta, SGTF infections diagnosed between November 2021 through January 2022 were classified as BA.1, SGTP infections diagnosed from February through April 2022 were classified as BA.2 and SGTF infections diagnosed in April 2022 were classified as BA.4/BA.5.

We used multivariable logistic regression models to assess risk factors for (i) hospitalisation and (ii) severe disease among hospitalised individuals (subset of individuals in model i), comparing Delta, BA.2, BA.4 and BA.5 infections to BA.1. We repeated the analysis comparing Omicron lineage infections to Delta. We controlled for factors associated with hospitalisation (age, sex, presence of co-morbidity, province and healthcare sector) and factors associated with severity (age, presence of co-morbidity, sex, province, healthcare sector, number of days between the dates of specimen collection and hospital admission and SARS-CoV-2 vaccination status) based on previously described predictors of outcome in South Africa[17,20]. Data on co-morbidities and reported SARS-CoV-2 vaccination were only available for hospitalised patients. To allow for at least three weeks of follow up, cases were censored to those with a specimen collected before 27 April 2022. Severity analysis was restricted to admissions that had already accumulated outcomes and all patients still in hospital were excluded.

We performed a sensitivity analysis, by using the same multivariable logistic regression models described above but restricting the time period to March and April 2022 (when BA.2 and BA.4/BA.5 were circulating), while adjusting for epidemiological week of infection in order to reduce bias due to differences in prior infection and vaccine-derived immunity, as well as changes in testing practices over time.

Severe disease was defined as a hospitalised patient meeting at least one of the following criteria: admitted to the intensive care unit (ICU), received any level of oxygen treatment, ventilated, received extracorporeal membrane oxygenation (ECMO), experienced acute respiratory distress syndrome and/or died, based on a modification of World Health Organization recommendations[21]. Co-morbidity was defined as ≥1 of the following conditions: hypertension, diabetes, chronic cardiac disease, chronic kidney disease, asthma, chronic obstructive pulmonary disease, malignancy, HIV, and active or past tuberculosis. Re-infection was defined as an individual with previous positive tests >90 days prior to the current episode from the SARS-CoV-2 laboratory test dataset. SARS-CoV-2 vaccination was defined as at least one dose of Ad26.COV2.S vaccine or at least two doses BNT162b. For variables where the proportion of missing data was small (<5%), an "unknown" category was not included in the categorisation of the variable. However, for vaccination, where a large proportion of the data was missing (55% missing), we included a category for "unknown". Multivariable analyses only included individuals with data available for all co-variates in the model, and therefore to avoid excluding a large proportion of individuals with missing vaccine status we included an unknown category for this variable.

Analysis was performed using Stata 14.1® (StataCorp LP, College Station, US). Categorical variables were summarised using frequency distributions and compared using Pearson's Chi-squared test. Pairwise interactions were assessed by inclusion of product terms for all variables remaining in the final multivariable additive models.

Ethical approval was obtained from the Human Research Ethics Committee (Medical) of University of the Witwatersrand for the collection of COVID-19 case and test data as part of essential communicable disease surveillance (M210752), and for the DATCOV surveillance programme (M2010108). The research conducted in this study is locally relevant, and included local researchers throughout the research process. Local and regional research relevant to this study was taken into account in the citations.

## Reporting summary

Further information on research design is available in the Nature Research Reporting Summary linked to this article.

## Data availability

The data generated and analysed during the current study contain potentially identifiable information and were shared with the national public health institute under the Notifiable Medical Conditions (NMC) regulations, and therefore have restricted access due to privacy and ethical issues. Access to aggregated data can be obtained by request to the corresponding author, Nicole Wolter (nicolew@nicd.ac.za), and will be subject to proof of an IRB-approved protocol and signature of a data sharing agreement. Responses to requests will be within three weeks from request receipt.

## Code availability

The code generated and used during the current study have been deposited and are available in a public repository[22].

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

## Acknowledgements

We acknowledge all NGS-SA members and laboratory teams, laboratory teams at the Centre for Respiratory Diseases and Meningitis and the Sequencing Core Facility of the NICD (Johannesburg, South Africa) for genomic sequencing data; and the national SARS-CoV-2 NICD surveillance team and NICD Information Technology team for NMCSS case data. We thank all laboratories for submitting specimens for sequencing, all public and private laboratories for COVID-19 diagnostic test data, and all public laboratories and Lancet Laboratories for ThermoFisher TaqPath™ COVID-19 PCR data. We thank all hospitals and healthcare workers submitting data through the DATCOV surveillance programme.

This study was funded by the South African Medical Research Council with funds received from the National Department of Health (CC). Sequencing activities for NICD are supported by a conditional grant from the South African National Department of Health as part of the emergency COVID-19 response; a cooperative agreement between the National Institute for Communicable Diseases of the National Health Laboratory Service and the United States Centers for Disease Control and Prevention (grant number 5 U01IP001048-05-00, CC); the African Society of Laboratory Medicine (ASLM) and Africa Centers for Disease Control and Prevention through a sub-award from the Bill and Melinda Gates Foundation grant number INV-018978 (AvG); the UK Foreign, Commonwealth and Development Office and Wellcome (Grant no 221003/Z/20/Z, CC); and the UK Department of Health and Social Care and managed by the Fleming Fund and performed under the auspices of the SEQAFRICA project (AvG). The Fleming Fund is a £265 million UK aid programme supporting up to 24 low- and middle-income countries (LMICs) generate, share and use data on antimicrobial resistance (AMR) and works in partnership with Mott MacDonald, the Management Agent for the Country and Regional Grants and Fellowship Programme. This research was also supported by The Coronavirus Aid, Relief, and Economic Security Act (CARES ACT) through the Centers for Disease Control and Prevention (CDC) and the COVID International Task Force (ITF) funds through the CDC under the terms of a subcontract with the African Field Epidemiology Network (AFENET) AF-NICD-001/2021 (AvG). Screening for SGTF at UCT was supported by the Wellcome Centre for Infectious Diseases Research in Africa (CIDRI-Africa), which is supported by core funding from the Wellcome Trust (203135/Z/16/Z and 222754).

The findings and conclusions in this manuscript are those of the author(s) and do not necessarily represent the official position of the funding agencies.

The funders played no role in the writing of the manuscript or the decision to submit for publication.

## Author contributions

Conception and design of study: N.W., W.J., S.W., A.vG., C.C. Data collection and laboratory processing: N.W., W.J., S.W., R.W., H.M., D.G.A., J.E., J.N.B., C.S., N.T., N.C., M.dP., N.G., A.I., A.G., K.M., W.S., F.K.T., K.S., Z.M., N.H., R.P., J.W., A.vG., C.C. Analysis and interpretation: N.W., W.J., S.W., R.W., H.M., M.G., D.G.A., J.E., J.N.B., C.S., N.C., M.dP., N.G., A.I., A.G., K.M., W.S., F.K.T., Z.M., N.H., R.P., J.W., H.H., M.D., A.B., A.vG., C.C. Accessed and verified the underlying data: N.W., R.W., H.M., D.G.A., J.E., A.vG. Drafted the Article: N.W., A.vG., C.C. All authors critically reviewed the Article.

## Competing interests

C.C. has received grant support from South African Medical Research Council, UK Foreign, Commonwealth and Development Office and Wellcome Trust, US Centers for Disease Control and Prevention and Sanofi Pasteur. NW has received grant support from Sanofi Pasteur and the Bill and Melinda Gates Foundation. AvG has received grant support from US Centers for Disease Control and Prevention, Africa Centres for Disease Control and Prevention, African Society for Laboratory Medicine (ASLM), South African Medical Research Council, WHO AFRO, The Fleming Fund and Wellcome Trust. SW has received grant support from US Centers for Disease Control and Prevention. RW declares personal shareholding in the following companies: Adcock Ingram Holdings Ltd, Dischem Pharmacies Ltd, Discovery Ltd, Netcare Ltd, Aspen Pharmacare Holdings Ltd. All other authors declare no conflict of interest.

## Additional information

[1]Centre for Respiratory Diseases and Meningitis, National Institute for Communicable Diseases (NICD) of the National Health Laboratory Service, Johannesburg, South Africa. [2]School of Pathology, Faculty of Health Sciences, University of the Witwatersrand, Johannesburg, South Africa. [3]Division of Public Health Surveillance and Response, National Institute for Communicable Diseases (NICD) of the National Health Laboratory Service, Johannesburg, South Africa. [4]Right to Care, Pretoria, South Africa. [5]School of Public Health, Faculty of Health Sciences, University of the Witwatersrand, Johannesburg, South Africa. [6]Centre for Tuberculosis, National Institute for Communicable Diseases (NICD) of the National Health Laboratory Service, Johannesburg, South Africa. [7]School of Health Sciences, College of Health Sciences, University of KwaZulu-Natal, KwaZulu-Natal, South Africa. [8]Centre for HIV and STIs, National Institute for Communicable Diseases of the National Health Laboratory Service, Johannesburg, South Africa. [9]SA MRC Antibody Immunity Research Unit, School of Pathology, Faculty of Health Sciences, University of the Witwatersrand, Johannesburg, South Africa. [10]Department of Biochemistry and Microbiology, Faculty of Science, Engineering and Agriculture, University of Venda, Thohoyandou, South Africa. [11]Sequencing Core Facility, National Institute for Communicable Diseases of the National Health Laboratory Service, Johannesburg, South Africa. [12]Lancet Laboratories, Johannesburg, South Africa. [13]National Health Laboratory Service (NHLS), Johannesburg, South Africa. [14]School of Laboratory Medicine and Medical Sciences, University of KwaZulu Natal, Durban, South Africa. [15]Division of Medical Virology, University of Cape Town, Cape Town, South Africa. [16]Department of Virology, University of KwaZulu-Natal, Durban, South Africa. [17]Department of Clinical Microbiology & Infectious Diseases, CH Baragwanath Academic Hospital, Johannesburg, South Africa. [18]Western Cape Government: Health and Wellness, and School of Public Health and Family Medicine, University of Cape Town, Cape Town, South Africa. [19]These authors contributed equally: Anne von Gottberg, Cheryl Cohen. ✉e-mail: nicolew@nicd.ac.za

