## [Peer Review File · Nature Communications]

Clinical severity of SARS-CoV-2 Omicron BA.4 and BA.5 lineages compared to BA.1 and Delta in South AfricaREVIEWER COMMENTS

Reviewer #1 (Remarks to the Author):

In this study, the authors investigated clinical severity of SARS-CoV-2 Omicron BA.4/BA.5 lineages in South Africa. It was found that SARS-CoV-2 Omicron BA.4/BA.5 show reduced clinical severity compared to previous variants including Omicron BA.1. While the data is interesting, and quite timely given the recent surges in BA.5 cases worldwide, this reviewer is concerned about the duplicate publication. The authors recently published a paper entitled "Outcomes of laboratory-confirmed SARS-CoV-2 infection in the Omicron-driven fourth wave compared with previous waves in the Western Cape Province, South Africa" *Trop Med Int Health*. 2022 May 10. While the earlier work was just from the western Cape region and the data herein are a much larger sample across multiple sites, the results in the second reprint are similar to those in this study.

The work was generally well considered and the analyses appropriate for the type of real world experientially evaluations that are necessary for such data.

Reviewer #2 (Remarks to the Author):

The manuscript "Clinical severity of SARS-CoV-2 Omicron BA.4 and BA.5 lineages in South Africa" by Wolter et al. is very well written and sound structured. It is a consistent adaption of the prior comparison of severity of BA.1 and BA.2 earlier published as preprint [doi:10.1101/2022.02.17.22271030], which allows for an early assessment of BA4/BA.5 clinical severity. The analysis of the recent variants BA.5 and BA.4 dominant in South Africa but also in large parts of the world allows for an early assessment of its clinical severity based on data linkage from surveillance system, laboratory data and clinical information for hospitalized patients. The authors apply an state-of-the-art statistical approach using multivariable logistic regression models to estimate the differences between the outcome of the infections with the different variants considering important epidemiological and clinical co-factors. These are described in the text and listed in tables. Current and relevant literature is listed and regarded in the manuscript.

The authors introduce the basic characteristics of the new variants BA.4 and BA.5 and refer to early publications regarding Delta, BA.1 and BA.2. The key tool of this analysis is, that observation period October 2021 to April 2022 is divided into phases based on the particular dominance of a variant. This diversion is very straight forward and founded. Although, the overlapping phase, of BA.1 and BA.2 in January 2022 could be discussed since a clear dominance of a variant (BA.1 or BA.2) could not be determined. Variant dominance was assessed using TaqPath assays, which detect S-Gene Target Positivity (SGTP) or failure (SGTF). SGTP and SGTF can be exploited, since the dominating variants still inexplicable alternate in the presence of the deletion at the position 69/70 in the Spike protein.

Authors discuss, that analysis is restricted to infections diagnosed using TaqPath, while technology might not be applied to the same extent throughout the country. Here it could help to show, how the number of TaqPath detected infections per province relates to the otherwise detected SARS-CoV-2 incidences and probably also hospitalizations in the provinces in the same time periods based on surveillance data.

However, as explained above for January 2022, which marks the take-over from BA.1 to BA.2 as dominant variant the results from the TaqPath show a quite even distribution of 55 % (BA.1) vs. 43 % (BA.2), see Supplementary Figure 1 and Supplementary Table 1. January 2022 accounts for about 17 % of all cases included in the analysis. Although, the main focus of the manuscript is not on BA.1 and BA.2 I suggest to elaborate on this, as the presented differences in the outcome of the analysis between BA.4/BA.5 and BA.1 and BA.2 are explained comparably equal.

To assess and compare the severity of the variants authors focus on hospitalization status and so-called disease severity of the cases. Disease severity is defined as hospitalization plus one additional criteria, which refers to intensive pulmonary care. However, regarding the hospitalization status it would be additionally very important to differentiate patients that were

admitted due to COVID-19 related symptoms or diagnosis and those that were tested positive for SARS-CoV-2 along with a not-COVID-19-related reason for hospitalization. In the footnote c from Table 1 authors state the patients were included as hospitalized, when admitted to hospital 7 days prior until 21 days after diagnosis. Especially for those cases admitted before diagnosis it is unclear if these patients were admitted with COVID-19 related symptoms. A number of not primarily COVID-19 cases admitted to hospitals could falsely increase hospitalization rate and severity assessment.

The results of the analysis are described compact to an sufficient extend. It remains a bit unclear, why authors use the age group of 13-18 as reference group within the multivariable logistic regression analysis for all patients (Table 2), while the age group 19-24 was used as reference for the analysis for disease severity among hospitalized cases (Table 3). I suggest to harmonize this.

Within the results section authors indicated the largest group of cases by > 25 years. Thereby, it remains a bit unclear, how this number was chosen as threshold. I would suggest to describe clearly all presented age groups or call median age for all patients. The description based on the specific age groups would also include the assumed a priori vulnerability. Especially considering the presence of risk factors and vaccination status which would be in line with the statement the authors made in the corresponding publication [doi:10.1101/2022.02.17.22271030] on BA.1 and BA.2 severity, that very young children and elderly above 60 showed the highest aORs. Additionally, this would include a reference to the vaccination status which in general is assumed to be varying between the age groups. Authors present the limited data on the vaccination status for the included case, i.e. only for hospitalised cases. Hereby, it is critical to report cases with one dose admission of BNT162b (2 dose std regime) together with one dose of Ad26.COVS.2 (1 dose std regime) as vaccinated.

Interestingly, the authors discuss the estimation of high population immunity by infection and vaccination and its impact on the overall measured clinical severity of BA.4/BA.5. Which is reported as similar to BA.1. Hereby, I would suggest to not speak about the clinical severity of individuals, rather than about the group of BA.1 vs. BA.4/BA.5 infected, because it might depend to the individual vaccination and immune status, which could not determined for a large fraction of cases. Finally, it is not in contrast to the finding of Kimura et al. [doi:10.1101/2022.05.26.493539] that the measure severity does not reflect the findings from the hamster study, and at least partially can be explained by the the national seroprevalence data. Although, it should be clearly mentioned in the discussion, that presented data of the fifth wave (BA.4/BA.5) is mainly from the first part of the wave, which might affect the younger and healthy population groups first, before it enters the more isolated and more vulnerable older age groups and thus also shifts severe outcome to the end, as it was observed in others countries before. If this is not the case in South Africa it could be shown for Delta, BA.1 and BA.2 with an age groups specific incidence comparison over time.

Authors close the discussion with the statement "Despite the emergence of BA.4/BA.5 leading to a fifth resurgence of cases in the country, this did not translate into severity levels observed in waves prior to Omicron." As this finding is clearly supported by the current and presented data I suggest to indicate in this sentence, that the study does cover the full fifth wave partially only and due to the so far low number of included cases (compared to BA.1 and BA.2) and the ongoing spread of BA.4/BA.5 it is probably too early for a final statement.

Overall, the manuscript substantially contributes to the current understanding of BA.4/BA.5 severity and the impact of the variants in South Africa. Key findings can help to assess the situation in other regions and methods description is detailed to allow for adaptation and supposedly reimplementation of analysis if described data is made available.

Kind regards,
Stefan Kröger

Reviewer #3 (Remarks to the Author):

Manuscript Review "Clinical severity of SARS-CoV-2 Omicron BA.4 and BA.5 lineages in South Africa"

Summary: In this manuscript, the authors assessed severity of SARS-CoV-2 infection, defined by hospitalization and severe disease once hospitalized, between Delta, BA.1, BA.2 and BA.4/5 variants. This manuscript is well-written and clear and has a highly relevant public health question. However, I do have some concerns regarding completeness of data and transparency in missing data presentation, modeling hospitalization with key covariates missing, and whether the approach sufficiently explores potential biases.

1.) It would be helpful to expand Supplementary Table 1 some to include additional data on the correspondence of SGT status and lineage for each month. Does the 86% for Oct 2021 indicate that 86% of samples that were SGTP were Delta? Or that 86% of overall samples were SGTP and were Delta? In January 2022, 55% of SGTF are BA.1, does this mean that 45% of SGTF are a different lineage? What is that lineage? This would have implications for the potential for bias in using a "predominance" threshold that is very permissive (i.e. 50% or more). Along these lines, please define "predominance" in line 114, is it 51%? If this is the case, the authors may consider omitting samples from January 2022 to enhance confidence in lineage designation. However, there is potential for misinterpretation of this table.

2.) (Lines 110-111): Please provide additional insight regarding the use of the TaqPath assay in this population. Does use vary by private vs. public sector? By region? By severity of infection? (i.e. for hospital, ED, vs. ambulatory swabs). Further, It seems that there is substantial variation in use during the time period and would be helpful to describe the impetus for these changes (Supplementary Figure 1). Given that there is a relatively small number of samples tested with Taqpath (16%), these data can help the reader understand the generalizability of the findings.

3.) Along these lines, it would be helpful to comment in the discussion what the potential role of home-based antigen testing is during this period. Are cases that are more severe more likely to be tested by PCR? Are PCR tests more commonly performed in healthcare settings which may overrepresent those being screened for procedures or other healthcare needs biasing testing to a sicker population?

4.) Please check Line 163 for a typo, do you mean "24.7% BA.2" instead of "24.7% BA.1"? If not, this sentence is confusing as written.

5.) BMI is not included as a comorbidity but has been widely shown to be associated with more severe disease from SARS-CoV-2 infection. Is BMI available? If not, this is a limitation that should be discussed.

6.) Table 1 does not clearly present missing data. For example, there should be a row for missing data for comorbidity and vaccination status rather than a footnote. All categories in a covariate should add to the column total.

7.) It is not clear how missing data are handled in analyses. For example, vaccination status is only available for 45% of hospitalized individuals. How were missing data for this covariate handled in the analyses for severe outcome? Was comorbid status available for all hospitalized patients? What about other variables?

8.) Prior vaccination and comorbid status are not included as covariates in the model for hospitalization as an outcome. It is a stretch to expect valid modeling of risk for hospitalization without these covariates as adjustment factors. They are clearly related to the outcome and also variant. Without these, I am concerned about my confidence in the analyses with hospitalization as an outcome.

9.) Even for those with self-reported vaccination status (45%), the data seem potentially unreliable and with very low uptake (29%). This is particularly in the context of the fact that prior to BA.1 vaccination in the South African population was 73%. Please explain reason for this discrepancy and should possibly be discussed in limitations.

10.) If seroprevalence data demonstrate that 97% of blood donors in S. Africa had SARS-CoV-2 antibodies post-BA.1, I am concerned that reinfection proportions in this study range from 3% to 12%? Modeling a variable with such a high likelihood of misclassification may have implications. While the authors do state likely under ascertainment of prior infection, there is little description of how they expect this to impact findings.

11.) Lines 197-198: Why do the authors state that odds of severe disease was lower for those with prior infection when the finding is completely null? There are other places in the manuscript where non-significant findings are reported as no difference.

12.) (Lines 249-250): it is stated that testing during BA.4/5 shifted to preferential testing among hospitalized individuals, and that this would make the estimate minimum. Please explain this further. Which estimate? Which model? What is the rationale behind this statement?

13.) Table 1. There appears to be a dramatic shift in cases over time and variant to predominate in the private sector. Do the authors have any thoughts as to why this is?

RESPONSE TO REVIEWER COMMENTS

Manuscript number: NCOMMS-22-25310

Manuscript title: Clinical severity of SARS-CoV-2 Omicron BA.4 and BA.5 lineages compared to BA.1 and Delta in South Africa

Reviewer #1:

In this study, the authors investigated clinical severity of SARS-CoV-2 Omicron BA.4/BA.5 lineages in South Africa. It was found that SARS-CoV-2 Omicron BA.4/BA.5 show reduced clinical severity compared to previous variants including Omicron BA.1. While the data is interesting, and quite timely given the recent surges in BA.5 cases worldwide, this reviewer is concerned about the duplicate publication. The authors recently published a paper entitled “Outcomes of laboratory-confirmed SARS-CoV-2 infection in the Omicron-driven fourth wave compared with previous waves in the Western Cape Province, South Africa” Trop Med Int Health. 2022 May 10. While the earlier work was just from the western Cape region and the data herein are a much larger sample across multiple sites, the results in the second reprint are similar to those in this study.

Response:

The paper described above (Trop Med Int Health. 2022) compares COVID-19 outcomes in the fourth wave (driven by Omicron BA.1) compared to previous waves in South Africa. Therefore, it compares Omicron BA.1 to Delta and Beta variants as well as to the ancestral virus. In contrast, our paper currently submitted differs from the Trop Med Int Health 2022 paper in that we analysed the severity of the fifth (BA.4/BA.5) wave compared to the fourth (BA.1) and third (Delta) waves. The Trop Med Int Health 2022 paper contains no data on BA.4/BA.5. In addition, the Trop Med Int Health 2022 paper was only conducted among adults in the public sector in one province in South Africa. This differs from the current paper in that we analysed the severity of the fifth (BA.4/BA.5) wave compared to the fourth (BA.1) and third (Delta) waves among individuals of all ages in both the public and private sector nationally.

The work was generally well considered and the analyses appropriate for the type of real world experientially evaluations that are necessary for such data.

Thank you for the positive feedback.

Reviewer #2:

Please note that we have also responded directly to reviewer 2 comments that were made in comments in the manuscript and have attached these files with the responses.

The manuscript “Clinical severity of SARS-CoV-2 Omicron BA.4 and BA.5 lineages in South Africa” by Wolter et al. is very well written and sound structured. It is a consistent adaption of the prior comparison of severity of BA.1 and BA.2 earlier published as preprint [doi:10.1101/2022.02.17.22271030], which allows for an early assessment of BA4/BA.5 clinical severity. The analysis of the recent variants BA.5 and BA.4 dominant in South Africa but also in large parts of the world allows for an early assessment of its clinical severity based on data linkage from surveillance system, laboratory data and clinical information for hospitalized patients. The authors apply an state-of-the-art statistical approach using multivariable logistic regression models to

estimate the differences between the outcome of the infections with the different variants considering important epidemiological and clinical co-factors. These are described in the text and listed in tables. Current and relevant literature is listed and regarded in the manuscript.

The authors introduce the basic characteristics of the new variants BA.4 and BA.5 and refer to early publications regarding Delta, BA.1 and BA.2. The key tool of this analysis is, that observation period October 2021 to April 2022 is divided into phases based on the particular dominance of a variant. This diversion is very straight forward and founded.

Although, the overlapping phase, of BA.1 and BA.2 in January 2022 could be discussed since a clear dominance of a variant (BA.1 or BA.2) could not be determined.

Response: We have added the following sentence to the introduction (lines 79-80) "In week 1 of 2022 BA.1 comprised the majority of sequences (72%), however was replaced by BA.2 during the first few weeks of 2022 and constituted 66% of sequences in week 4." Even though these two Omicron lineages were co-circulating in January 2022, we were able to distinguish between infections with these two lineages in our analyses as BA.1 is SGTF and BA.2 is SGTP on the TaqPath assay.

Variant dominance was assessed using TaqPath assays, which detect S-Gene Target Positivity (SGTP) or failure (SGTF). SGTP and SGTF can be exploited, since the dominating variants still inexplicable alternate in the presence of the deletion at the position 69/70 in the Spike protein.

Authors discuss, that analysis is restricted to infections diagnosed using TaqPath, while technology might not be applied to the same extent throughout the country. Here it could help to show, how the number of TaqPath detected infections per province relates to the otherwise detected SARS-CoV-2 incidences and probably also hospitalizations in the provinces in the same time periods based on surveillance data.

Response: We have added this data as suggested in Supplementary table 2, and added the following to the results section (lines 170-174) "In the period 1 October 2021 through 26 April 2022, 884,379 COVID-19 cases were diagnosed. Of these, 144,086 (16.3%) were known to have been diagnosed using the TaqPath™ COVID-19 PCR test (Supplementary figure 1), ranging by province from 3.7% to 23.6% of cases (Supplementary table 2). In the public sector 11.1% (43752/395924) of cases were diagnosed with the TaqPath test, and 20.5% (100334/488455) in the private sector (P<0.001)."

The limitation of this data has been discussed in the limitations paragraph (lines 278-280) as follows: "The analysis is restricted to infections that were diagnosed using the TaqPath™ COVID-19 PCR assay, which was not used to the same extent throughout the country and therefore may bias the data geographically."

However, as explained above for January 2022, which marks the take-over from BA.1 to BA.2 as dominant variant the results from the TaqPath show a quite even distribution of 55 % (BA.1) vs. 43 % (BA.2), see Supplementary Figure 1 and Supplementary Table 1. January 2022 accounts for about 17 % of all cases included in the analysis. Although, the main focus of the manuscript is not on BA.1 and BA.2 I suggest to elaborate on this, as the presented differences in the outcome of the analysis between BA.4/BA.5 and BA.1 and BA.2 are explained comparably equal.

Response: As mentioned above, we have added the following sentence to the introduction (lines 79-80) “In week 1 of 2022 BA.1 comprised the majority of sequences (72%), however was replaced by BA.2 during the first few weeks of 2022 and constituted 66% of sequences in week 4.”

In addition, we included the following in the discussion (lines 261-268): “Among the group of individuals with BA.2 infection, we observed a slightly reduced risk of hospitalisation and severe disease compared to those infected with BA.1, although bordering on statistical significance. We previously reported that a similar proportion of individuals were hospitalised and developed severe illness for individuals infected with BA.1 compared to BA.2, although at an earlier point towards the end of the fourth wave in January 2022⁷. This may be because the BA.2 lineage became dominant in the period when SARS-CoV-2 case numbers were declining from the fourth wave (BA.1) peak and population immunity was high with high numbers of infections and re-infections during this wave¹⁸.”

To assess and compare the severity of the variants authors focus on hospitalization status and so-called disease severity of the cases. Disease severity is defined as hospitalization plus one additional criteria, which refers to intensive pulmonary care. However, regarding the hospitalization status it would be additionally very important to differentiate patients that were admitted due to COVID-19 related symptoms or diagnosis and those that were tested positive for SARS-CoV-2 along with a not-COVID-19-related reason for hospitalization. In the footnote c from Table 1 authors state the patients were included as hospitalized, when admitted to hospital 7 days prior until 21 days after diagnosis. Especially for those cases admitted before diagnosis it is unclear if these patients were admitted with COVID-19 related symptoms. A number of not primarily COVID-19 cases admitted to hospitals could falsely increase hospitalization rate and severity assessment.

Response: The DATCOV hospitalisation surveillance system contains data for all individuals with a positive PCR or antigen test, with a confirmed duration of stay in hospital of ≥ 1 day, regardless of age or reason for admission. We are therefore not able to distinguish between patients with COVID-19 related symptoms and those with other reasons for admission with incidental SARS-CoV-2 diagnosis. We included individuals that were admitted to hospital between 7 days prior to 21 days after diagnosis (specimen collection date) because individuals may only have been tested and diagnosed once admitted to hospital. This group may also include individuals that acquired SARS-CoV-2 infection nosocomially.

We have added the following to the methods section (lines 116-120): “DATCOV contains data for all individuals with a positive PCR or antigen test, with a confirmed duration of stay in hospital of ≥ 1 day, regardless of age or reason for admission. This included patients with COVID-19 symptoms, acquired nosocomial COVID-19 infection, or tested positive incidentally when admitted for other reasons.”

In addition, we have included the following in the limitations section of the discussion (lines 309-313): “Lastly, DATCOV surveillance includes individuals with a SARS-CoV-2 positive test that may have been hospitalised with COVID-19 symptoms or for other non-COVID-19 related conditions which may have overestimated the number of hospitalisations. However, the DATCOV case definition was consistent throughout the study period, and would have affected each time period in the analysis consistently.”

The results of the analysis are described compact to an sufficient extend. It remains a bit unclear, why authors use the age group of 13-18 as reference group within the multivariable logistic

regression analysis for all patients (Table 2), while the age group 19-24 was used as reference for the analysis for disease severity among hospitalized cases (Table 3). I suggest to harmonize this.

Response: Thank you for alerting us to this. The age group 19-24 was used as the reference group in both analyses. The data was incorrectly entered in Table 2, and has been corrected to reflect the correct reference age group. The correct reference group was used in the text of the results section.

Within the results section authors indicated the largest group of cases by > 25 years. Thereby, it remains a bit unclear, how this number was chosen as threshold. I would suggest to describe clearly all presented age groups or call median age for all patients. The description based on the specific age groups would also include the assumed a priori vulnerability. Especially considering the presence of risk factors and vaccination status which would be in line with the statement the authors made in the corresponding publication [doi:10.1101/2022.02.17.22271030] on BA.1 and BA.2 severity, that very young children and elderly above 60 showed the highest aORs.

Response: Thank you for this suggestion. We have included in results the median age of cases by variant as follows (lines 179-181): "The median age and interquartile range of cases for each variant was 37 (25-52) years, 37 (28-49) years, 35 (17-48) years and 38 (27-51) years for Delta, BA.1, BA.2 and BA.4/BA.5 respectively." We have removed the description of age of cases using a threshold of >25 years as suggested.

Additionally, this would include a reference to the vaccination status which in general is assumed to be varying between the age groups. Authors present the limited data on the vaccination status for the included case, i.e. only for hospitalised cases. Hereby, it is critical to report cases with one dose admission of BNT162b (2 dose std regime) together with one dose of Ad26.COVS.2.S (1 dose std regime) as vaccinated.

Response: During the time period of this study in South Africa, the COVID-19 vaccine guidelines did not allow for mixed vaccine schedules, therefore individuals would have received either BNT162b or Ad26.COVS.2.S. In our analysis (previously only indicated in the footnote to the tables) we defined vaccination as at least one dose of SARS-CoV-2 vaccine (either BNT162b or Ad26.COVS.2.S), as suggested by the reviewer. This has been made clearer by including the following sentence in the methods (lines 156-157): "Vaccination was defined as at least one dose of SARS-CoV-2 vaccine (either BNT162b or Ad26.COVS.2.S)."

We have added COVID-19 vaccine coverage in the South African population in the introduction as follows (lines 84-87): "As of end April 2022, 49.6% of individuals aged ≥ 18 years in South Africa had received at least one dose of COVID-19 vaccine (BNT162b or Ad26.COVS.2.S): 36.2%, 53.4%, 65.7% and 70.7% in the 18-34 years, 35-49 years, 50-59 years and ≥ 60 years age groups respectively."

Interestingly, the authors discuss the estimation of high population immunity by infection and vaccination and its impact on the overall measured clinical severity of BA.4/BA.5. Which is reported as similar to BA.1. Hereby, I would suggest to not speak about the clinical severity of individuals, rather than about the group of BA.1 vs. BA.4/BA.5 infected, because it might depend to the individual vaccination and immune status, which could not be determined for a large fraction of cases.

Response: The discussion has been updated as follows (lines 246-247): "In our study, among the group of individuals infected with BA.4/BA.5 lineages we did not observe increased clinical severity compared to the group of individuals infected with BA.1."

Finally, it is not in contrast to the finding of Kimura et al. [doi:10.1101/2022.05.26.493539] that the measure severity does not reflect the findings from the hamster study, and at least partially can be explained by the the national seroprevalence data.

Although, it should be clearly mentioned in the discussion, that presented data of the fifth wave (BA.4/BA.5) is mainly from the first part of the wave, which might affect the younger and healthy population groups first, before it enters the more isolated and more vulnerable older age groups and thus also shifts severe outcome to the end, as it was observed in others countries before. If this is not the case in South Africa it could be shown for Delta, BA.1 and BA.2 with an age groups specificity incidence comparison over time.

Response: The following was stated in the limitations paragraph of the discussion (lines 301-304): “In addition, we compared infections through the full Omicron BA.1 wave to infections in the ascending phase of the Omicron BA.4/BA.5 wave, this could bias comparisons if case characteristics differ in the ascending and descending wave phases or if threshold for hospitalisation changed in these time periods.”

As suggested by the reviewer we have added the following to the limitations (lines 304-307): “While the earlier phase of a wave may affect the younger and healthy population groups first before reaching the more vulnerable populations, previous data from DATCOV has indicated the proportion of severe cases does not vary substantially through the different wave periods²⁰.”

Authors close the discussion with the statement “Despite the emergence of BA.4/BA.5 leading to a fifth resurgence of cases in the country, this did not translate into severity levels observed in waves prior to Omicron.” As this finding is clearly supported by the current and presented data I suggest to indicated in this sentence, that the study does cover the full fifth wave partially only and due to the so far low number of included cases (compared to BA.1 and BA.2) and the ongoing spread of BA.4/BA.4 it is probably too early for a final statement.

Response: We agree with the reviewer and have adapted this sentence in order to soften the wording (lines 317-320): “Despite the emergence of BA.4/BA.5 leading to a fifth resurgence of cases in the country, data from early in the wave indicate that this may not translate into severity levels observed in waves prior to Omicron when population immunity was lower.”

Overall, the manuscript substantially contributes to the current understanding of BA.4/BA.5 severity and the impact of the variants in South Africa. Key findings can help to assess the situation in other regions and methods description is detailed to allow for adaption and supposedly reimplementation of analysis if described data is made available.

Thank you for the feedback.

Kind regards,
Stefan Kröger

Reviewer #3:

Manuscript Review “Clinical severity of SARS-CoV-2 Omicron BA.4 and BA.5 lineages in South Africa”
Summary: In this manuscript, the authors assessed severity of SARS-CoV-2 infection, defined by hospitalization and severe disease once hospitalized, between Delta, BA.1, BA.2 and BA.4/5 variants. This manuscript is well-written and clear and has a highly relevant public health question. However, I

do have some concerns regarding completeness of data and transparency in missing data presentation, modeling hospitalization with key covariates missing, and whether the approach sufficiently explores potential biases.

1.) It would be helpful to expand Supplementary Table 1 some to include additional data on the correspondence of SGT status and lineage for each month. Does the 86% for Oct 2021 indicate that 86% of samples that were SGTP were Delta? Or that 86% of overall samples were SGTP and were Delta? In January 2022, 55% of SGTF are BA.1, does this mean that 45% of SGTF are a different lineage? What is that lineage? This would have implications for the potential for bias in using a “predominance” threshold that is very permissive (i.e. 50% or more). Along these lines, please define “predominance” in line 114, is it 51%? If this is the case, the authors may consider omitting samples from January 2022 to enhance confidence in lineage designation. However, there is potential for misinterpretation of this table.

Response: Supplementary table 2 has been expanded to more clearly indicate the distribution of variants by month based on genomic surveillance data. We have included a column for “Other” so that the proportion of variant based on genomic data by month is easier to interpret. In addition, we have included the proportion of TaqPath tests that were SGTF and SGTP by month to show this correlation with the genomic sequence data.

We agree that the use of the term “predominance” was not clear, and we have therefore removed this term and updated the wording of this sentence as follows (lines 122-126): “We used a combination of the presence (S-gene target positive, SGTP) or absence (S-gene target failure, SGTF) of the S-gene together with time period of circulating variants/lineages from genomic surveillance in South Africa (Supplementary table 1), as a proxy for the variant and lineage.”

2.) (Lines 110-111): Please provide additional insight regarding the use of the TaqPath assay in this population. Does use vary by private vs. public sector? By region? By severity of infection? (i.e. for hospital, ED, vs. ambulatory swabs). Further, It seems that there is substantial variation in use during the time period and would be helpful to describe the impetus for these changes (Supplementary Figure 1). Given that there is a relatively small number of samples tested with Taqpath (16%), these data can help the reader understand the generalizability of the findings.

Response: A range of PCR assays are used in both public and private laboratories in South Africa, depending on the platforms available and availability of PCR reagents. We have added Supplementary table 2 which shows the proportion and variability of cases diagnosed with the TaqPath PCR test by province. We have added the following to results (lines 173-174): “In the public sector 11.1% (43752/395924) of cases were diagnosed with the TaqPath test, and 20.5% (100334/488455) in the private sector”. We unfortunately do not have data on whether the swabs were taken in hospital, ED or ambulatory and are therefore not able to describe whether there was differential use of the TaqPath assay in these settings. However, in South Africa the test used is largely based on test availability at the laboratory and not by patient severity.

3.) Along these lines, it would be helpful to comment in the discussion what the potential role of home-based antigen testing is during this period. Are cases that are more severe more likely to be tested by PCR? Are PCR tests more commonly performed in healthcare settings which may

overrepresent those being screened for procedures or other healthcare needs biasing testing to a sicker population?

Response: As of July 2022, home-based antigen testing is not yet available in South Africa. Both antigen and PCR testing are offered and performed by healthcare workers in South Africa. It is possible that sicker individuals in healthcare settings were more likely to have PCR tests and therefore bias the population in this study. However, as stated by the reviewer, this would have resulted in bias towards more severe disease. We have added the following to the discussion (lines 280-283): “Additionally, it is possible that individuals with more severe disease were more likely to have been diagnosed by PCR than antigen tests, resulting in potential bias of our study population towards more severe disease.”

4.) Please check Line 163 for a typo, do you mean “24.7% BA.2” instead of “24.7% BA.1”? If not, this sentence is confusing as written.

Response: This was a typo, thank you for picking it up. We have corrected to 24.7% BA.2.

5.) BMI is not included as a comorbidity but has been widely shown to be associated with more severe disease from SARS-CoV-2 infection. Is BMI available? If not, this is a limitation that should be discussed.

Response: Although BMI was included for hospitalised patients in the DATCOV questionnaire, this data was missing for 98% of cases and therefore could not be included in the analyses. We have added the following to the limitations (lines 307-309): “We did not have data on body mass index, a known risk factor for severe COVID-19, and could therefore not include this variable in our analysis.”

6.) Table 1 does not clearly present missing data. For example, there should be a row for missing data for comorbidity and vaccination status rather than a footnote. All categories in a covariate should add to the column total.

Response: In order to simplify the tables, we did not include missing/unknown categories for the co-variates. However, for each covariate we indicated separately the total number (denominator) of individuals for which data was available in the table (N=xx). This would not affect the multivariable analyses as only individuals with complete data across all co-variates in the model are included.

7.) It is not clear how missing data are handled in analyses. For example, vaccination status is only available for 45% of hospitalized individuals. How were missing data for this covariate handled in the analyses for severe outcome? Was comorbid status available for all hospitalized patients? What about other variables?

Response: We have added the following to methods (lines 157-162): “For variables where the proportion of missing data was small (<5%), and “unknown” category was not included in the categorisation of the variable. However, for vaccination, where a large proportion of the data was missing (55% missing), we included a category for “unknown”. Multivariable analyses only included individuals with data available for all co-variates in the model, and therefore to avoid excluding a large proportion of individuals with missing vaccine status we included an unknown category for this variable.”

8.) Prior vaccination and comorbid status are not included as covariates in the model for hospitalization as an outcome. It is a stretch to expect valid modeling of risk for hospitalization without these covariates as adjustment factors. They are clearly related to the outcome and also variant. Without these, I am concerned about my confidence in the analyses with hospitalization as an outcome.

Response: We agree, but unfortunately this data was only collected for hospitalised patients through the DATCOV surveillance programme. We have included this as a limitation as follows (lines 287-289): "Vaccination information was restricted to hospitalised cases and was based on self-report, and as a result the analysis of severe disease among hospitalised individuals was likely more robust than the hospitalisation analysis."

9.) Even for those with self-reported vaccination status (45%), the data seem potentially unreliable and with very low uptake (29%). This is particularly in the context of the fact that prior to BA.1 vaccination in the South African population was 73%. Please explain reason for this discrepancy and should possibly be discussed in limitations.

Response: At the end of April 2022, 49.6% of individuals aged ≥ 18 years had received at least one dose of vaccine. This has been added to the introduction of the manuscript (lines 84-87). This is similar to the 52.0% of hospitalised individuals that reported having been vaccinated during the fifth wave (BA.4/BA.5) of infections (Table 1).

10.) If seroprevalence data demonstrate that 97% of blood donors in S. Africa had SARS-CoV-2 antibodies post-BA.1, I am concerned that reinfection proportions in this study range from 3% to 12%? Modeling a variable with such a high likelihood of misclassification may have implications. While the authors do state likely under ascertainment of prior infection, there is little description of how they expect this to impact findings.

Response: We agree with the reviewer that our adjustment for reinfections was unlikely to have fully accounted for the effect of previous infection. Using our data-linkage approach in a comprehensive dataset of all diagnosed and reported cases in South Africa it is likely that we did identify the vast majority of previously diagnosed infections. However, as mentioned in the discussion, "less than 10% of SARS-CoV-2 infections are diagnosed" in South Africa. What this would mean is that we were not able to adequately adjust for the effects of previous infection in the multivariable models and therefore some of the observed effect of lower severity in the BA.4/BA.5 wave compared to pre-Omicron waves may be as a result of immunity from prior infection rather than reduced intrinsic virulence. Similarly, if a substantially higher proportion of individuals infected with BA.4/BA.5 had undiagnosed infection compared to BA.1 this could lead to a false impression of equal severity when in fact the intrinsic severity of BA.4/BA.5 could be somewhat higher than that of BA.1. This is indicated in the discussion in lines 289-296.

11.) Lines 197-198: Why do the authors state that odds of severe disease was lower for those with prior infection when the finding is completely null? There are other places in the manuscript where non-significant findings are reported as no difference.

Response: This sentence has been updated as follows (lines 218-221): "The odds of severe disease did not differ for individuals with prior infection (aOR 0.9, 95% CI 0.6-1.3)."

12.) (Lines 249-250): it is stated that testing during BA.4/5 shifted to preferential testing among hospitalized individuals, and that this would make the estimate minimum. Please explain this further. Which estimate? Which model? What is the rationale behind this statement?

Response: In the same way that PCR testing may be biased towards individuals with severe disease, any testing may have this same effect. If testing guidelines changed during the BA.4/BA.5 period to only recommend testing (PCR or antigen) for hospitalised individuals, then our study population would be biased towards more severe disease, resulting in our estimate of risk of hospitalisation for BA.4/BA.5 infected individuals being an underestimate. We have updated this sentence in the limitations for clarity as follows (lines 298-301): “During the BA.4/BA.5 wave in some provinces there was a shift to preferential testing of hospitalised individuals, which would have biased our study population of individuals tested for SARS-CoV-2 to more severe disease and would therefore make our estimate for risk of hospitalisation of BA.4/BA.5 infected individuals a minimum estimate.”

13.) Table 1. There appears to be a dramatic shift in cases over time and variant to predominate in the private sector. Do the authors have any thoughts as to why this is?

Response: It is not clear why there is this trend towards the private sector in later waves, but may be influenced by a number of differences between the public and private sectors in South Africa. It may be due to reduced COVID-19 testing among individuals and clinicians in the public sector as testing protocols shifted to hospitalised individuals, whereas individuals in the private sector would have had consistent access to testing even when the need for COVID-19 diagnosis for public health interventions for containment became less essential. This has been discussed in lines (270-276).

REVIEWER COMMENTS

Reviewer #2 (Remarks to the Author):

The resubmitted manuscript "Clinical severity of SARS-CoV-2 Omicron BA.4 and BA.5 lineages in South Africa" by Wolter et al. is very well written and has a clear structure. It is a consistent adaptation of the prior comparison of severity of BA.1 and BA.2 earlier published as preprint [doi:10.1101/2022.02.17.22271030], which allows for an early assessment of BA4/BA.5 clinical severity. The authors addressed questions and comments of the reviewer and added important details and substantial data and information to the manuscript. As for example the divergence of TaqPath usage throughout the country within the observation period was described and further explained compared to the prior submitted version.

However, some concerns remain, which is i) the analysis of hospitalization data without differentiating between COVID- and non-COVID-related admission. ii) the inclusion of one-dose BNT and one dose Ad26.COVS.2 cases with the set of vaccinated cases and iii) the regression analysis including self-reported reinfection data that the authors describe as under-ascertained.

A remaining limitation concerns the included hospitalization data. The authors explain that specific data on the reason for hospitalization is not available within the DATCOV surveillance data set. It remains problematic, that no differentiation can be made between those cases hospitalized for COVID-19 relevant symptoms and cases that are admitted to hospitals for non-COVID-19 related conditions and were tested positive for SARS-CoV-2. Although the authors describe the problem, they do not explain the rationale, that this bias is consistent over the study period and why the bias can be assumed as constant over time (Line 309:313).

As prior mentioned, it is disputable to report cases with one dose admission of BNT162b (2 dose std regime) together with one dose of Ad26.COVS (1 dose std regime) as vaccinated – implicating fully vaccinated. Although the authors included a specific statement in the methods section, to explain why the pooling of cases either receiving one dose of BNT162b or Ad26.COVS was not adapted/resolved. Still, one dose of BNT162b should not be reported as fully vaccinated, although it may provide comparable protection for a very limited period, while for Ad26.COVS one dose relates to the full vaccine regimen. If the vaccination guidelines in South Africa in the study period differ in that categorization, it should be pointed out and referenced. The effect of a full vaccination (as defined for approval procedure) regarding disease severity especially for hospitalized cases can be assumed as different. The effectiveness of full vaccination has been shown in different studies, e.g. Nyberg et al. 2022 ([https://doi.org/10.1016/S0140-6736\(22\)00462-7](https://doi.org/10.1016/S0140-6736(22)00462-7)); Harder et al. 2021 (<https://doi.org/10.2807/1560-7917.ES.2021.26.41.2100920>), Thomas et al. 2021 (<https://www.nejm.org/doi/10.1056/NEJMoa2110345>), Tang et al. 2021 (<https://doi.org/10.1038/s41591-021-01583-4>).

Also, the authors report that South African population has reached a very high level of seroprevalence (up to 97 %) within the study period. The additional benefit of single dose BNT vaccination, is difficult to interpret. Therefore I recommend to refer to the group of fully vaccinated cases (2-dose BNT or 1 dose Ad26.COVS) only and to consequently adapt the multivariable logistic regression analysis.

In line with the interpretation of vaccination data, the validity of the reinfection data and the derived interpretation is questionable. Because the reported percentage of reinfection (3-12 %) stands in huge contrast to the number of 87 % seroprevalence by prior infection, the authors report with reference to Bingham et al. 2022 and Pulliam et al. (both cited by the authors).

Although, authors discuss that only 10 % of SARS-CoV-2 infections are diagnosed, they use the estimated reinfection rate of 3-12% (from self-reporting) for regression analysis. The recently added sentence "The odds of severe disease did not differ for individuals with prior infection (aOR 0.9, 95% CI 0.6-1.3)." (lines 218-221) potentially misleads interpretation considering the limitations of the available reinfection data. The inclusion of reinfection into the regression analysis has no added value, as authors correctly discuss the high rate of high impact population immunity by prior infection and vaccination.

Reviewer #3 (Remarks to the Author):

no further comments. concerns were adequately addressed.

RESPONSE TO REVIEWER COMMENTS

Manuscript number: NCOMMS-22-25310

Manuscript title: Clinical severity of SARS-CoV-2 Omicron BA.4 and BA.5 lineages compared to BA.1 and Delta in South Africa

Reviewer #2 (Remarks to the Author):

The resubmitted manuscript “Clinical severity of SARS-CoV-2 Omicron BA.4 and BA.5 lineages in South Africa” by Wolter et al. is very well written and has a clear structure. It is a consistent adaption of the prior comparison of severity of BA.1 and BA.2 earlier published as preprint [doi:10.1101/2022.02.17.22271030], which allows for an early assessment of BA4/BA.5 clinical severity.

The authors addressed questions and comments of the reviewer and added important details and substantial data and information to the manuscript. As for example the divergence of TaqPath usage throughout the country within the observation period was described and further explained compared to the prior submitted version.

However, some concerns remain, which is i) the analysis of hospitalization data without differentiating between COVID- and non-COVID-related admission. ii) the inclusion of one-dose BNT and one dose Ad26.COVS.2 cases with the set of vaccinated cases and iii) the regression analysis including self-reported reinfection data that the authors describe as under-ascertained.

Thank you for the feedback. Please see responses to each of these concerns below.

A remaining limitation concerns the included hospitalization data. The authors explain that specific data on the reason for hospitalization is not available within the DATCOV surveillance data set. It remains problematic, that no differentiation can be made between those cases hospitalized for COVID-19 relevant symptoms and cases that are admitted to hospitals for non-COVID-19 related conditions and were tested positive for SARS-CoV-2. Although the authors describe the problem, they do not explain the rationale, that this bias is consistent over the study period and why the bias can be assumed as constant over time (Line 309:313).

We have added the following to explain the reason for the inability to differentiate between these groups in the DATCOV dataset as well as data to support that it did not change over time in the discussion (page 11, lines 308-313): “While DATCOV surveillance contains a field in the web-based platform to indicate if a person was admitted for COVID-19 symptoms or for another reason, the submitted data for this field was often incomplete. As a result, data on reason for admission is missing for approximately 60% of patients. Among those for whom data was available, the proportion of patients admitted for COVID-19 symptoms was 75% (first wave), 78% (second wave), 76% (third wave), 70% (fourth wave) and 74% (fifth wave). In addition, the DATCOV case definition was consistent throughout the study period, and would have affected each time period in the analysis consistently.”

As prior mentioned, it is disputable to report cases with one dose admission of BNT162b (2 dose std regime) together with one dose of Ad26.COVS (1 dose std regime) as vaccinated – implicating fully vaccinated. Although the authors included a specific statement in the methods section, to explain

why the pooling of cases either receiving one dose of BNT162b or Ad26.COVS.2 was not adapted/resolved. Still, one dose of BNT162b should not be reported as fully vaccinated, although it may provide comparable protection for a very limited period, while for Ad26.COVS.2 one dose relates to the full vaccine regimen. If the vaccination guidelines in South Africa in the study period differ in that categorization, it should be pointed out and referenced. The effect of a full vaccination (as defined for approval procedure) regarding disease severity especially for hospitalized cases can be assumed as different. The effectiveness of full vaccination has been shown in different studies, e.g.

Nyberg et al. 2022 ([https://doi.org/10.1016/S0140-6736\(22\)00462-7](https://doi.org/10.1016/S0140-6736(22)00462-7)); Harder et al. 2021 (<https://doi.org/10.2807/1560-7917.ES.2021.26.41.2100920>), Thomas et al. 2021 (<https://www.nejm.org/doi/10.1056/NEJMoa2110345>), Tang et al. 2021 (<https://doi.org/10.1038/s41591-021-01583-4>).

Also, the authors report that South African population has reached a very high level of seroprevalence (up to 97 %) within the study period. The additional benefit of single dose BNT vaccination, is difficult to interpret. Therefore I recommend to refer to the group of fully vaccinated cases (2-dose BNT or 1 dose Ad26.COVS.2) only and to consequently adapt the multivariable logistic regression analysis.

As suggested by the reviewer we have updated the manuscript to define vaccinated individuals as having received at least one dose of Ad26.COVS.2 vaccine or at least two doses of BNT162b vaccines. Changes in this regard have been made to the following sections of the manuscript including the multivariable logistic regression analysis for risk of severe disease.

1. Methods: page 6, lines 152-153
2. Results: page 7, lines 183-187; page 8, lines 206-222
3. Table 1: page 17
4. Table 3: page 20-21

In line with the interpretation of vaccination data, the validity of the reinfection data and the derived interpretation is questionable. Because the reported percentage of reinfection (3-12 %) stands in huge contrast to the number of 87 % seroprevalence by prior infection, the authors report with reference to Bingham et al. 2022 and Pulliam et al. (both cited by the authors).

Although, authors discuss that only 10 % of SARS-CoV-2 infections are diagnosed, they use the estimated reinfection rate of 3-12% (from self-reporting) for regression analysis. The recently added sentence “The odds of severe disease did not differ for individuals with prior infection (aOR 0.9, 95% CI 0.6-1.3).” (lines 218-221) potentially misleads interpretation considering the limitations of the available reinfection data. The inclusion of reinfection into the regression analysis has no added value, as authors correctly discuss the high rate of high impact population immunity by prior infection and vaccination.

Re-infection rates were not self-reported but based on the identification of an individual with at least one positive SARS-CoV-2 test >90 days prior to the current episode from the SARS-CoV-2 laboratory test dataset. This has been clarified in the methods (page 6, lines 150-151) as follows: “Re-infection was defined as an individual with previous positive tests >90 days prior to the current episode from the SARS-CoV-2 laboratory test dataset”

The odds of hospitalisation and severe disease for BA.4/5 infections compared to BA.1 infections did not differ when comparing multivariable models including and excluding re-infections (see below table). As suggested by the reviewer we have removed reinfections from the multivariable models.

In this regard the manuscript has been updated in the following places:

1. Methods, lines 131-133
2. Results, lines 215-216
3. Discussion, lines 285-288
4. Table 2: pages 18-19
5. Table 3: pages 20-21

Model	Including re-infection	Excluding re-infection
	aOR (95%CI)	aOR (95%CI)
Risk of hospitalization	1.24 (0.98-1.55)	1.24 (0.98-1.55)
Risk of severe disease	0.72 (0.41-1.27)	0.72 (0.41-1.26)

Reviewer #3 (Remarks to the Author):

no further comments. concerns were adequately addressed.

Thank you for the feedback.